# Follicular Dynamics and Pregnancy Rates during Foal Heat in Colombian Paso Fino Mares Bred under Permanent Grazing

**DOI:** 10.3390/ani14050760

**Published:** 2024-02-29

**Authors:** Mauricio Cardona-García, Claudia Jiménez-Escobar, María S. Ferrer, Juan G. Maldonado-Estrada

**Affiliations:** 1OHVRI-Research Group, Faculty of Agrarian Sciences, College of Veterinary Medicine, University of Antioquia, Medellín 050034, Colombia; mauricio.cardonag@udea.edu.co; 2Grupo de Investigación en Reproducción Animal y Salud de Hato, Facultad de Medicina Veterinaria y Zootecnia, Universidad Nacional de Colombia, Bogotá 111321, Colombia; cjimeneze@unal.edu.co; 3College of Veterinary Medicine, The University of Georgia, Athens, GA 30602, USA; msferrer@uga.edu

**Keywords:** edema score, dominant follicle, foal heat, postpartum ovulation, pregnancy rate, preovulatory follicle, uterine edema

## Abstract

**Simple Summary:**

A high percentage of mares (92–94%) exhibit foal heat in the first 21 days postpartum, and insemination during this estrus is critical to ensure a higher fertility rate per breeding season. A retrospective descriptive study evaluated follicular dynamics peripartum and at foal heat, uterine edema, and endometrial cytology between mares that did or did not become pregnant when inseminated. Mares became pregnant if inseminated when the foal heat started from the second postpartum week, had a lower growth rate of the larger diameter follicle and had an adequate degree of edema compared to non-pregnant mares.

**Abstract:**

No studies have evaluated the peripartum follicular dynamics resulting in foal heat under tropical environments. We aimed to assess retrospectively the peripartum follicular dynamics in Colombian Paso Fino mares that were inseminated at the foal heat, becoming pregnant or not. Records including follicular dynamics of pregnant mares prepartum and from foaling until foal heat ovulation were assessed in Colombian Paso Fino mares (CPF, n = 24) bred under permanent grazing in a tropical herd in Colombia. The number of ovarian follicles >10 mm before foaling and the largest follicle (F1) growth rate (mm/day) from foaling until the F1 reached the largest diameter (pre-ovulatory size) at the foal heat were assessed. Mares were inseminated at foal heat with 20 mL of semen (at least 500 million live spermatozoa) with >75% motility and 80% viability from a stallion of proven fertility. Ovulation was confirmed the day after follicles had reached the largest diameter. Quantitative data from follicular growth, the day at ovulation, from mares that became pregnant (PM) or not (NPM) at 16 days post-insemination were compared by one-way ANOVA, repeated measures ANOVA (follicle growth rate data) or Chi-square test (edema and cytology scores data). Epidemiological data, gestation length, and the number of follicles on third prepartum days did not significantly differ between PM and NPM (*p* > 0.05). Seventy-one percent of mares (17/24) got pregnant. Ovulatory follicles grew faster in the NPM group (n = 7), which ovulated between the seventh and ninth postpartum days, compared to PM (n = 17), which ovulated between the 11th and 13th postpartum days. Pre-ovulatory follicle diameter in PM (48.57 ± 0.8 mm) was significantly larger than in NPM (42.99 ± 1.0 mm) (*p* < 0.05). In addition, the PM edema score (2.93 ± 0.32 mm) on ovulation day was significantly lower (*p* < 0.05) than NPM (4.47 ± 0.05 mm). First postpartum ovulation occurred at 12.6 ± 0.3 and 8.5 ± 0.4 days (*p* < 0.05) in PM and NPM, respectively. Colombian Paso Fino mares bred under permanent grazing under tropical rainforest conditions with no foaling or postpartum complications showed a 71% gestation rate when inseminated at foal heat when ovulation occurs between the second and third postpartum week.

## 1. Introduction

The equine industry worldwide requires mares to foal each year during the foaling season [1,2], a goal that could not be achieved if the mare does not get pregnant at the foal heat because the mare’s gestation length lasts 340 days on average [1]. This fact allows a narrow margin for decisions in case of impaired fertility during the foal heat, meaning mares have only a putative window of 26 days for becoming pregnant after foaling, which results in 16 to 18 days if considering uterine involution. Mares must exert a coordinated uterine involution with early resumption of postpartum follicular dynamics to resume postpartum cyclicity, present the foal’s heat, and become pregnant. This fact calls for constantly advancing knowledge of the factors influencing the foal heat, its duration variability, and the resulting pregnancy rate. In equine reproductive practice in Colombia, mares are bred in foal heat by natural mating or artificial insemination, mainly if no foaling and postpartum complications occur. Authors recommend breeding mares in foal heat advanced uterine involution as shown by endometrial cytological exam and ovulation at or after the tenth postpartum day [3,4,5]. There are reports on follicular dynamics and pregnancy rates after foal heat [6,7,8,9], but no information is available on these processes in Colombian Creole mares reared under grazing conditions. Equine practitioners, however, routinely perform breeding at foal heat throughout the country in Colombian Paso Fino mares [10,11].

Improving the probability of mares becoming pregnant if bred at the foal heat is a permanent challenge in equine reproductive practice and reproductive biotechnology [12]. Several factors can influence pregnancy rates of the foal heat, including housing conditions (e.g., grazing versus stall housing) [13], breed, time of year [14], time of placental delivery [15], and the persistence of inflammation and delayed uterine involution related to obstetric interventions at foaling, which could alter uterine involution histological pattern [16]. Furthermore, the pregnancy rate may vary depending on whether the new embryo is implanted in the previous gestation’s pregnant versus no pregnant uterine horn [17], the mare’s postpartum body condition score [18], and the number of postpartum heats [19] or early postpartum days at breeding [20].

Whether postpartum therapeutic interventions affect pregnancy rates in mares bred at the foal heat has been reported. So, uterine lavage performed in between two and four postpartum days did not affect the presence of uterine bacteria, uterine inflammation, or pregnancy rate of mares bred in the foal heat [21]. If the first postpartum ovulation occurs early when uterine involution is not completed, the pregnancy rate of mares is impaired if inseminated at the foal heat. Accordingly, a four-year retrospective study including the postpartum follow-up of Quarter Horse (n = 838) and Thoroughbred (n = 939) mares showed that mares mated at the foal heat showed a significantly lower pregnancy rate compared with mares bred at subsequent estrus following the foal heat [22]. Another report involving 99 Thoroughbred eutocia-foaled mares inseminated at the foal heat revealed that 18.2% had free intrauterine fluid during foal heat. These mares showed increased embryo mortality between 15 and 45 days of gestation than mares having no uterine fluid [23]. In Colombia, insemination in the foal heat is routinely practiced in most equine breeding systems, in mares bred in stables [24], or under permanent grazing systems [25]. It has been accepted that the gestation rates of mares bred in the foal heat are low compared to the second postpartum heat [18]. In this context, in a recent study by Gastal et al. 2022 performed in postpartum mares, the authors found a shorter interval from foaling to ovulation during the foal heat, a larger dominant follicle (F1), and an earlier day at which the F1 reached the maximum diameter during the foal heat [26]. Moreover, breeding the mare at the second postpartum heat involves a risk of the mare entering postpartum anestrus related to metabolic and seasonal issues [1,27], with the resumption of ovarian cyclicity only after weaning.

Interestingly, reducing stressful conditions reduced embryo death in mares bred at the foal heat compared to when stressful conditions were not controlled [28]. Then, research is needed to provide objective information for a more rational approach to inseminating mares in foal heat. The pregnancy rate after foal heat breeding depends on the day estrus begins: 1—early, at the end of the first week; 2—normal, between the second and third week; and 3—late, after the third week [29]. Reports support the relationship between the first postpartum ovulation and pregnancy rates in Thoroughbred mares [30,31]. 

To decide if it is recommended to inseminate the mare in the foal heat, factors such as ease of foaling, postpartum complications [32], uterine involution [31], and foaling date [33] must be considered. Due to scarce information on the follicular dynamics of the foal heat in Colombian Creole mares, studies are needed to set up the best timing for insemination or mating under tropical conditions. Defining more precise clinical criteria to decide if inseminating the mare in the foal heat is a growing demand of horse owners to maximize pregnancy rates and pregnancy maintenance. 

Studies undertaken in mares from breeds other than CPF provided insights on the follicular dynamics and estrous cycle endocrinology in the mare; affecting factors including high altitude [11], endocrine factors and their interaction [2,34], and factors affecting follicular dynamics of the foal heat including postpartum uterine involution [4], the effect of placental retention [15], gestation rates according to the previous pregnant uterine horn [17], season and parity [31], age of the mare [35], postpartum first mating schedules [36]; and in donkeys [8], age [20], breed, and season [37]. Sharma et al. [31], found that 93% of the mares presented foal heat in the first 21 days postpartum in India, with no statistically significant effect of variables such as stud farm, region, time of year, number of mating, or parity [20,31]. Until now, few of these variables have been shown in Colombian Paso Fino mares bred under permanent grazing conditions in low-altitude tropics. With the working hypothesis that follicular dynamics preceding the first postpartum heat affect pregnancy rate in mares inseminated during the foal heat, the objective of the present study was to study prepartum and postpartum follicular dynamics in Colombian Creole mares and to evaluate how they affect the time to the first postpartum ovulation and pregnancy rate in foal heat.

## 2. Materials and Methods

### 2.1. Study Design

A descriptive-retrospective study was performed using recorded data of Colombian Paso Fino (CPF, n = 14) and CPF crosses with Quarter Horses (QH, n = 10) that became or were not pregnant after receiving artificial insemination at the foal heat. The average (±SD) age at foaling was 6.6 ± 1.3 and 7.5 ± 1.2 years for CPF and QH mares, respectively. Foaling occurred from October 2018 to January 2019 and from October 2019 to January 2020. The mares had been followed until June 2020. Mares were raised and bred under permanent grazing in two private farms (farm one, and farm two), located in the municipality of Puerto Berrío, in the State-like region of Antioquia (Colombia). The altitude ranges between 50 and 125 m above sea level, corresponding to the life zone of tropical rainforest. Both farms were in the Magdalena Medio tropical region of Antioquia, Colombia, 10 km from each other in a region that has 28 °C (minimal 27 °C, maximum 30 °C) annual average temperature and 2400 mm/year average annual rainfall [38]. Mares were permanently fed on grass pastures of *Dichanthium annulatum* (Kleberg’s bluestem) pastures mixed with native grasses (Figure 1). The two farms were chosen for convenience because they belonged to the same company, had adequate mares routinely checked during gestation, parturition, and puerperium, and were raised under the same environmental, nutritional, and management conditions. The mares had been born, raised, and remained on these two farms for at least 20 years at the time of data analysis. The mares were supplemented with mineralized salt having 6% phosphorus, a standard concentration of trace and macro elements, and water ad libitum. Mares received no concentrated food. In farm one, they managed the purebred CPF mares and crossbred CPF × QH mares, while in farm two, they managed only the purebred CPF. No statistically significant differences in epidemiological data were found between farms. 

The farms did not record data on the bromatological or foliar analysis of forages or soil analysis. The rainiest months correspond to the May-September season, while the driest months correspond to the November-April season [30]. The mares were born, raised, and bred in the same herd, and they foaled in good general health without any gestation or foaling complications. All foals survived, and all mares were lactated throughout the study.

### 2.2. Inclusion and Exclusion Criteria

Inclusion criteria. 1—Pregnant Mares aged 5 to 10 years old. 2—Pure-bred Colombian Paso Fino (CPF) or cross-bred Quarter Horse x CPF genetic background. 3—Mares bred by artificial insemination. 4—Mares fed climacuna Angleton grass (*Dichanthium annulatum*) and native grasses, receiving mineralized salt holding 6% phosphorus and water ad libitum. 5—No clinical signs indicative of systemic disease. 6—Body condition score ranging from 6 to 7 (1 to 9 scale) at foaling. 7—No history of dystocia in the previous or current foaling. 8—No postpartum complications.

Exclusion criteria: Exclusion criteria. 1—Mares out of normal health condition. 2—Mares with a history of previous abortions. 3—Mares above or below the established BCS (less than five or greater than 7 (on a scale of 1 to 9). 4—Mares with retained fetal membranes. And 5—Mares with dystocia.

### 2.3. Reproductive Follow-Up

Ultrasonographic evaluation of ovarian activity had been started at 300 days of gestation on mares in which it was possible to reach the ovaries for rectal palpation and ultrasound evaluation. These exams were performed every three days until foaling (day 0). Data analysis comprised the number of follicles > 10 mm at days −3 and 0 (foaling day), the diameter of the largest (F1) and second largest (F2) follicles at day 0, and postpartum days 5, 7, 9, 10, 11, 12, and 13. Also, data on the degree of uterine edema around the periovulatory days and the day of artificial insemination at the foal heat were recorded. When the F1 follicle reached 40 to 45 mm, it was considered a candidate for ovulation. Then, the mare was evaluated twice or three times daily (every 6 h from 06:00 to 18:00). When the follicle started to become deformed, it was considered a pre-ovulatory follicle, and the mare was inseminated with fresh semen from a fertile stallion. The next reproductive exam was performed 12 h later, and the absence of the previously diagnosed F1 follicle was used as a criterion to confirm ovulation. No ovulation-inducing agents were used. For follicles greater than 45 mm, inseminations were repeated every 48 h until ovulation was confirmed. Pregnancy was evidenced by ultrasound evaluation on days 16, 30, and 40 after insemination. According to pregnancy status, mares were grouped into pregnant (PM) and non-pregnant (NPM), and the retrospective analysis of the data from follicular dynamics, first postpartum ovulation, endometrial cytology, and uterine edema (see below for the corresponding procedure) were analyzed and compared between PM and NPM. 

### 2.4. Foals’ Inspection

Clinical inspection of the foals: Newborn foals were evaluated by clinical inspection immediately postpartum early in the morning (between 5:30 and 6:00 a.m.) according to the herd’s routine (all foaling had occurred overnight without supervision). Foals were evaluated by visual inspection, allowing us to verify they had stood as confirmed by the presence of meconium in the perianal region. They were nursed as confirmed by vigor in moving the liveliness and alertness of the eyes. It was assumed that there was no dystocia because the foals had normal respiratory movements, with no signs of dyspnea or cyanotic mucous membranes to suggest any respiratory distress. The foal’s umbilicus was also evaluated by visual inspection and palpation, and the umbilical cord was dipped in a 5% iodine solution to prevent infectious or fly infestation.

### 2.5. Foaling Day and Postpartum Mares’ Assessment

Most foaling occurred between December and April (Table 1), considered a dry season in the region. Mares’ assessment was performed daily in the morning (6:00 to 7:00 h). Because there was no evidence of any abnormality in the foaled mares and the foals, it was assumed that mares foaled naturally with no dystocia. 

The mares were evaluated early in the morning, immediately after the foals’ assessment at the time indicated above. The mares foaled in a 10 ha paddock at dawn, which impeded placental examination. Therefore, placenta expulsion was assumed and evidenced by the absence of fetal membranes protruding from the vulva and the lack of fetal membrane remnants into the uterus, as assessed by a transrectal ultrasound exam. Those mares found in the foaling process were moved to the stable and seen until the placenta was inspected immediately after foaling when no pathological conditions were detected. The presence of any fluid content in the uterus and hyperechoic structures in the lumen of the uterus were evaluated with ultrasonographic examination. In addition, the day zero ovarian ultrasound exam was performed. No cases of postpartum metritis or any other puerperium abnormality were detected in the mares.

#### 2.5.1. Reproductive Ultrasound Exam

At each prepartum and postpartum examination, transrectal palpation was performed first, followed by an ultrasonographic examination to quantify the number of follicles and follicle diameter, using a Sonoscape A5 portable ultrasound scanner (Sonoscape Medical Corp., Shenzhen, Guangdong, China) equipped with a multi-frequency linear transducer from 5 to 7.5 MHz. These examinations were performed daily, starting at 300 days of gestation; once the foaling date was known, records from three days before and on the day of foaling were taken. Postpartum follow-up was performed at five and seven days, every day from nine to fourteen days, and then at 24 h after insemination to confirm ovulation. Once ovulation was verified, the follicular follow-up was stopped and followed by pregnancy diagnosis 14 and 40 days later by detecting the embryonic vesicle. Inseminations were performed when the largest follicle reached its maximum diameter before becoming irregular in shape, and the disappearance of the F1 follicle was assessed the previous day, followed by confirmation of the corpus hemorrhagic, verified ovulation, and the subsequent formation of the corpus luteum [2,34].

#### 2.5.2. Follicle Growth

Follicle diameter was recorded and calculated according to Newcomb’s [2] recommendation. Briefly, for each follicle, four measures were taken, imagining the 12 to 6, 3 to 9, 1:30 to 7:30, and 4:30 to 10:30 positions in a clock, using the electronic calipers of the ultrasound machine, and the average distance was recorded as the average follicle diameter. The follicles with the largest (F1) and second largest (F2) diameter were checked every 12 hours. Since the F1 follicle reached 40 mm, it was evaluated at 6:00, 12:00, and 18:00 h. When the follicle was found to be deformed, indicative of an ovulation process, insemination was performed, and the largest diameter reached in the evaluation before its deformation was considered the F1 pre-ovulatory diameter. The disappearance of the F1 follicle detected in the exam performed 12 h before and the finding of a CL formation assessed by ultrasound confirmed ovulation. Data on follicle growth were analyzed for mares that became pregnant or not pregnant according to the postpartum day of evaluation and ovulation (days 3, 5, 9, 10, 11, 12, or 13).

#### 2.5.3. Uterine Edema

Uterine edema was evaluated according to the guidelines by Newcomb [2] and graded from 1 to 5, where 1 means no folds, 3 means the typical folding patterns during estrus, and 5 means excessive/abnormal folding.

#### 2.5.4. Endometrial Cytology

Endometrial cytology samples were taken on days 7 and 9 postpartum and on the day when the follicle reached its largest diameter before becoming deformed, using a sterile cytobrush technique. The cytobrush was rolled onto glass slides stained with Diff-Quick and Gram stain. Under the microscope, cell morphology was assessed, and epithelial and inflammatory cells were counted. The uterine inflammatory status was classified as mild (5–10% PMN), moderate (10–20% PMN), or severe (>21% PMN), considering 5% PMN as a cut-off point [39,40].

#### 2.5.5. Semen Collection and Processing

A single stallion was always used as a semen donor, collected using a Hannover artificial vagina, with the water temperature at 45 °C, guaranteeing 38 °C internally. The semen was diluted to a final concentration of 50 × 10^6^ sperm/mL using a refrigeration semen extender. The insemination dose had 1250 × 10^6^ spermatozoa with progressive motility between 75% and 80%. The insemination was performed by conventional technique with deposition of the semen into the body of the uterus. 

### 2.6. Statistical Analysis

Data included for the analysis consisted of the number of previous foalings, age of the mare, gestation length, body condition at foaling, number of follicles before foaling, follicular sizes at foaling, and postpartum ovarian and uterine follow-up. During the foal heat, the degree of edema and the results of endometrial cytology on the day of ovulation and artificial insemination were evaluated. First, data were analyzed by descriptive statistics. Furthermore, the information obtained was compared between mares that did and did not become pregnant, using one-way ANOVA. The number of follicles larger than 10 mm was counted in each ovary. The diameter of the largest follicle was determined by averaging four dimensions obtained with the Sonoscape A5V ultrasound. The follicular growth rate was calculated by the difference between the diameter of the day of ovulation and the diameter of the day of the previous evaluation divided by the number of days between each measurement: e.g., five days for growth between foaling and day five postpartum; two days between postpartum days five and seven, and days seven and nine; and one day between days 9 and 14 postpartum. Comparison of prepartum, foaling, and postpartum follicular dynamics values between mares that became pregnant or non-pregnant when inseminated in foal heat was evaluated by a one-way ANOVA test. Follicle growth between pregnant and non-pregnant mares was assessed by repeated-measures ANOVA. The Chi-square test was used to evaluate differences in edema scores at insemination.

## 3. Results

Pre-partum data were available from 24 mares from which the ovaries were reached, and mares foaled between 2019 (n = 14) and 2020 (n = 10) could be imaged. There were no statistically significant differences in data between pure-bred or crossbred mares (*p* > 0.05), so the mares were evaluated as a whole group. Individual and grouped demographic data for age, weight, foaling date, and duration of gestation for each group of mares are presented in Table 1 and Table 2, respectively. None of the mares presented complications during or after foaling; no cases of retained placenta or injuries to the external genitalia resulting from foaling were found. A neonatal exam was performed within 30 to 60 min after birth. The parameters of the neonatal exam were within normal limits in all foals. 

### 3.1. Epidemiological Data

The mean (±SEM) values of the mare’s epidemiological variables are presented in Table 2. There were no statistically significant differences between PM and NPM. No statistically significant differences were found in body weight between Paso Fino mares and Paso Fino mares crossed with Quarter Horse mares (*p* = 0.4578), nor between Paso Fino mares that were subsequently pregnant and those that were not (*p* = 0.7336). It was impossible to compare with the crossbred mares that did not become pregnant because there were only two. No statistically significant differences (*p* > 0.05) were found for epidemiological variables between PM and NPM (Table 2).

### 3.2. Prepartum and Foaling Day Follicular Dynamics

The number of follicles smaller than 10 mm at three days prepartum and at foaling on each of the ovaries was not different between pregnant mares (PM) and non-pregnant mares (NPM) (Table 3). The diameter of the largest F1 follicle at foaling was significantly larger (*p* = 0.004) in the NPM group than in the PM group (Table 3).

### 3.3. Postpartum Follicular Dynamics

No double ovulation was observed in any of the mares. Pre-ovulatory follicle growth for the F1 and F2 follicles of each group of mares is shown in Figure 2. 

In the PM group, the mean (±SEM) diameter of the F1 follicle ranged from 12.1 ± 1.2 to 45.2 ± 1.8 mm in the PM group from the foaling to the 13th postpartum day and from 19.4 ± 2.1 to 49.5 ± 1.2 in the NPM group since the foaling to the 9th postpartum day (Table 4). In the PM group, the follicular growth rate was significantly higher from foaling to postpartum days 5 (*p* < 0.05), 7, 9, and 10 (*p* < 0.01). In NPM, the follicular growth rate was statistically significantly higher at all time points evaluated (Figure 2). Similarly, the F1 diameter was significantly higher (*p* < 0.01) in mares from the NPM group compared to the PM group (Table 4). The mean maximum ovulatory follicle diameter on each day of ovulation is presented in Table 5.

The average follicular growth rate (±SEM) was significantly higher in the NPM than in the PM group (*p* < 0.05), and ovulation during the foal heat occurred earlier in the NPM (8.5 ± 0.4 days) than in the PM group (12.6 ± 0.3 days) (*p* = 0.013) (Figure 3).

### 3.4. Uterine Findings

#### 3.4.1. Endometrial Cytology

Severe inflammatory cytology typical of the postpartum period remained at 100% and 43% of the NPM on postpartum days 7 and 9, respectively. Severe inflammation predominated on day five in 100% of the PM, while moderate and mild inflammatory cytology between 9 and 12 days postpartum was the predominant cytology finding in this group (Table 6).

#### 3.4.2. Uterine Edema

The urine edema score at the time of insemination was 2.44 + 0.2 and 4.84 + 0.1 for the PM and NPM groups, respectively. Uterine edema in NPM remained constant throughout estrus onset until ovulation (*p* > 0.05), while edema in PM showed a downward trend (Figure 4).

## 4. Discussion

Based on a retrospective descriptive data analysis, the present work aimed to study the follicular dynamics in mares bred under permanent grazing in a humid tropical farm from three days before foaling until the first postpartum ovulation occurring during the foal heat. Information available on follicular dynamics of the estrous cycle in the mare, its endocrinology, and factors affecting it [2,11,41], and on follicular dynamics of foal heat in donkeys [8,37] and mares [4,15,17,31,35,36] have not been evidenced in Colombian Paso Fino mares bred under permanent grazing conditions, as found in the present study. This study reports the follicular dynamics from three days before foaling, the day of foaling, and from the postpartum days 5 to 13 in CPF mares. The number of ovarian follicles smaller than 10 mm found at days −3 and foaling did not significantly differ between groups (*p* > 0.05). Interestingly, the NPM group showed follicular dynamics indicating significantly higher F1 and F2 follicles at foaling and postpartum day 5 (*p* < 0.01), which resulted in earlier initiation of the foal heat and the first postpartum ovulation (Figure 1). Furthermore, the F1 growth rate was statistically significantly higher than in the PM group, which resulted in all NPM groups having ovulated a postpartum day nine. On the contrary, mares of the PM group showed a prolonged follicular growth phase, with maximum growth reached at day ten and no significant differences between postpartum days 10 and 13 (Figure 1). Another interesting finding was that the F2 growth rate was significantly higher in the NPM group than in PM, with a maximum F2 diameter ranging from 20 to 30 mm between postpartum days five to nine (Figure 1). These results suggest that the NPM group started follicular recruitment before foaling and had more rapid growth rates than the PM group during the first week postpartum. So, the mean (± SEM) diameter of F1 in NPM at foaling was 19.4 (±2.1) mm, compared to 12.05 (±1.2) mm in the PM group (*p* < 0.01). These differences in follicular size and growth rate accounted for an early onset of the foal heat in the NPM group. This resulted in an earlier first postpartum ovulation at postpartum day 7 in NPMs related to pregnancy failure. 

The mean F1 size at foaling in NPM groups and early postpartum foal heat and ovulation suggests that follicle divergence could have occurred just before foaling, followed by a rapid growth period and early postpartum and ovulation. If considering as reference for F1 follicle deviation 22–25 mm diameter in mares [6,41], in our study, F1 deviation occurred between the second and third postpartum days in the NPM group and around eight days in the PM group, respectively (Figure 2). However, the significantly higher F1 diameter found earlier postpartum in NPM and PM groups suggests deviation occurred earlier at the foaling day when the F1 diameter was 15 and 10 mm in NPM and PM groups, respectively (Figure 2, Table 4). One can assume that this parameter could be related to the small size of CPF mares, whose mean body weight was 341.8 (±2.8) and 340.0 (±3.0) kg for mares of the PM and NPM groups, respectively. On the contrary, none of the factors considered could affect follicular dynamics (body weight at foaling, gestation length, parity, and body condition score) and significantly affect the results (*p* > 0.05).

In the report by Sharma et al. [31], the authors evaluated the follicular activity and reproductive performance in mares bred under subtropical conditions in India, where 93% of the mares presented foal heat in the first 21 days postpartum. In contrast, in our study, all mares exhibited foal heat between five and nine days postpartum, which resulted in the growth of follicles that reached pre-ovulatory diameter as early as seven days postpartum, as occurred in the NPM group. Sharma et al. [31] also reported that there was no statistically significant effect of the mares’ stud farm, region, season, number of mating, or parity but a lower proportion of older mares presenting the foal heat compared to younger mares [20,31].

In our study, mares foaled between December and May, corresponding to the rainy season in that region, where only a bimodal phase of the rainy and dry seasons occurs [38]. Because our research did not evaluate the climatic records, we cannot argue whether there was a seasonal effect. In seasonal countries, the season significantly affects follicular dynamics, increasing ovarian follicular activity and ovulation in spring and summer and suppressing it in fall and winter [18,36,37]. In our study, there was no effect of the epidemiological variables indicated in Table 2, supporting the idea that the mares were subjected to homogeneous management. The first postpartum ovulation in subtropical regions was reported on average at day 21 postpartum (n = 190 mares) [31], compared to our study where the mean postpartum day at ovulation was 8.5 and 12.6 days for NPM and PM groups, respectively.

One of the pioneering works classified postpartum follicular dynamics in mares as early (ovulation before the tenth) or late (ovulation after the tenth day postpartum) [29]. These authors reported that the number of follicles larger than 25 mm in the first three days postpartum was higher. The ovulatory follicle was more prominent in diameter in mares with early ovulation, unlike mares exhibiting late ovulation, who presented the highest number of follicles larger than 20 mm between 4- and 7 days post-foaling [29]. In our study, no statistical difference was found in the number of follicles smaller than 10 mm, but the F1 follicle was significantly larger at foaling in the NPM group than in the PM group. Interestingly, all F1 detected at foaling became the ovulatory follicle in both groups of mares. In the NPM group, three (43%) and four (57%) F1 follicles ovulated from the left and right ovary, respectively. In contrast, in the PM group, four (24%) and thirteen (76%) mares ovulated from the left and right ovary, respectively. In our study, all mares that presented early postpartum ovulation did not become pregnant, while all mares that showed late ovulation postpartum ovulation became pregnant, contrary to a report in which no statistically significant differences between early or late ovulation were found [29]. 

Regarding the type of housing and stressful conditions, it is currently accepted that grazing is better than stalls for horse raising and breeding. Furthermore, housing conditions related to the type of food and stress present risk factors that affect horses with stereotypies, which are worsened by stabling [42]. Accordingly, authors reported changes in physiological variables during estrus related to increased sympathetic adrenergic activity with behavioral variations towards conspecifics and humans. In contrast, physiological variables decreased at the end of estrus, indicating a decrease in sympathetic activity [25]. In addition, in a study in which the authors evaluated metabolites related to stress and inflammation in the serum of mares during the perinatal period, the authors concluded that under normal perinatal conditions where mares suffer no disturbances in health during pregnancy, foaling, and lactation, there could be improved and reproductive performance [43]. In our study, these factors are likewise reduced in free-ranging and permanent grazing systems.

In this context, the pregnancy rate in our study (70.8%) is higher than other pregnancy rates reported for the foal heat. Thoroughbred mares in which the embryo was implanted in the non-pregnant or pregnant horn of the previous gestation showed gestation rates of 79.2% and 20.8%, respectively, suggesting that uterine involution plays a critical role in early embryo implantation and development, as reported by Davies et al. [17]. The pregnancy rate in our study is similar to the report of a 66.7% gestation rate in Arabian mares in Algeria [44]. In a study conducted in Japan through a survey of 292 producers, the authors evaluated the foal heat performance of 1432 mares. The average (±standard deviation) placental expulsion time was 58 ± 88 min, with mares that had retained the placenta exhibiting lower gestation rates than mares lasting less than three hours to expel the placenta [15]. Our study found neither retained placenta nor foaling or puerperal complications. These findings could be due to the natural conditions under permanent grazing where the mares of our research were raised, in agreement with a report where the authors reported better uterine involution and higher pregnancy rates of mares bred under natural conditions than mares bred under stalls or higher manipulation conditions [13].

In a study conducted on Finn horse mares, a gestation rate of 45.7% was found in mares inseminated at foal heat, which ovulated on average at 12.5 ± 4.6 days postpartum (±standard deviation) [5], in agreement with our PM group data. Nevertheless, the authors reported an overall gestation rate of 94.3% when gestations resulting from breeding at the second and third postpartum heats were pooled. In our study, mares were inseminated at the foal heat. On the contrary, our results differed from reports where earlier or late postpartum ovulation affected reproductive performance in mares [42,45] and jennies [46,47]. Studies also reported lower pregnancy rates if mares were bred at the foal heat compared to the second postpartum heat [19,20]. The results of these reports could be related to stressful conditions [43], impaired uterine involution, or uterine infection. 

The age of the mares in our study, which ranged between 6.58 (pregnant group) and 7.1 years (non-pregnant group), did not significantly differ. On the contrary, authors reported lower gestation rates due to higher embryonic mortality in mares older than ten years bred at foal heat than mares younger than ten years [35].

In our study, pre-ovulatory follicles presented similar growth rates when comparing the results between days 5 and 9 (NPM) and days 9 and 11 (PM) (Figure 2). However, mares in the PM group had much higher growth rates on days 9–10 (8.3 ± 3.1) and 10 to 11 (6.2 ± 3.3), which were not observed in the NPM group. The F1 follicles of the NPM group showed a lower growth rate but ovulated faster than those in the PM group. Despite having performed no hormonal measurements, the ovulatory follicles of the NPM group could have presented an impaired endocrine profile, given that the edema score and inflammatory endometrial cytology were significantly higher than those of the PM group. An increased inflammatory profile related to uterine inflammation could alter estrogen production by the developing F1 follicle or the endometrial response to hormonal stimulus, resulting in impaired uterine reactions required to support implantation and early embryonic development. Furthermore, this could explain the finding of F2 follicles developing at diameters higher than those observed for the F2 follicles in the NP group (Figure 2, Table 4). 

Uterine edema found during insemination was 2.44 + 0.2 and 4.84 + 0.1. for PM and NPM groups, respectively (Figure 4). Could this be due to the severe inflammatory cytology at the time of insemination in the non-pregnant group (Table 6), the pathological pattern of edema that predominated in this group (Figure 4), or both? It could be a matter for further studies. Endometrial histology of postpartum pony mares has been reported, where the authors concluded that the expression of estrogen receptors allowed the endometrium to respond to estrogen during foal heat and to progesterone in subsequent diestrus [48]. 

Differences in pregnancy rates between early and late-ovulated mares could be related not only to the viability of the embryos, which must reach a suitable uterine environment for implantation [49], but also to uterine involution and PMN content [4]. Our study showed no evidence of postpartum alterations, so it is assumed that the mares could have had normal uterine involution. In a study conducted in Cold-Blood Polish Mares, in which prolactin, serum amyloid A, and biochemical analytes –(A.S.T., Ca^2+^, Mg^2+^, P+, T-Chol, and TP)- were measured during the peripartum, the authors found no significant statistical differences in the analytes evaluated between foaling and the foal heat [44]. 

Given that fetal cortisol levels have been reported in the Mare’s blood and saliva from several days before foaling up to foaling [50], we wonder whether some differences in the hormonal profile—sex steroids and adrenal steroids—could affect the endocrine processes responsible for follicular dynamics before and after foaling, in a degree enough to affect implantation rate [50]. This topic deserves further study. Finally, equine practitioners must consider a collective interpretation of data on an individual basis, including follicular size at foaling, postpartum day of the foal heat, significant follicle growth rate, uterine edema score, and cytology score, to decide whether to inseminate or not mares at the foal heat.

Progesterone measurements were not performed in the present work due to problems beyond the researchers’ control. In a study, the authors reported no statistically significant differences in progesterone levels between pregnant and non-pregnant mares at 13 days after ovulation [51]. Authors have concluded that progestogen levels in mares during foal heat are like those of cyclic mares. Specifically, 5α-di-hydro progesterone and allopregnanolone drop significantly from three days before foaling [52] to typical estrus concentrations at 48 h post foaling on average, such that, during foal heat and its corresponding ovulation, progesterone concentrations are like those of a cyclic mare during estrus [44]. Further studies could evaluate whether gestation failures in mares inseminated in foal heat could be related to alterations in the progestogen depletion profile between foaling and the first postpartum ovulation of the foal heat. In a study of mares receiving uterine lavages with high volumes of fluids between two and four days postpartum, no statistically significant difference was found in foal heat gestation rates (55.5% and 68.4%, respectively) [21], consistent with that reported by Blanchard et al. [53]. In our study, the gestation rate was higher when mares were not subjected to manipulation other than ultrasonographic follow-up and endometrial cytology.

What could explain the findings of follicular dynamics in our study? Firstly, the natural environment in which the mares have been raised and bred for at least 20 years implies they were adapted to the environmental, nutritional, and management conditions provided on the farms. Second, the routine reproductive exams could tell they were adapted to the management during consecutive reproductive exams, reducing the stress-related discomfort typical of mares bred in stables. The body condition score of mares at foaling and balanced nutritional support provided by permanent grazing warrants adequate nutritional reserve to support lactation, resumption of ovarian cyclicity, and fertile postpartum ovulation while lactating the foals [54,55,56]. Finally, differences in peripartum follicular dynamics between PM and NPM groups could be due to individual variations, as if the NPM group were more susceptible than the NP group to postpartum uterine inflammation as reported for cycling mares [57].

## 5. Conclusions

Our results provide evidence for equine practitioners on more accurate ways to decide on inseminating (or mating) mares during the foal heat under tropical conditions. The mares must be assured of a standard body condition score (range 6 to 7 on a 1 to 9 scale), normal foaling, and no postpartum complications. At foaling, the mares should be examined for the regular postpartum assessment. During the first postpartum week, ultrasound evaluation must be focused on ovarian follicular development and the growth rate of the F1 and F2 follicles. In our study, the F1 follicle shows a moderate growth rate, and the foal heat starts in the second or third postpartum week. In that case, mares’ assessment will result in ovulation between the second and third postpartum week, which, combined with adequate uterine edema and a mild to slight endometrial cytology, would result in high pregnancy rates. Specific mare-based reproductive follow-up must be guaranteed for mares with dystocia and postpartum complications, as well as for mares foaling with suboptimal health conditions.

## Figures and Tables

**Figure 1 animals-14-00760-f001:**
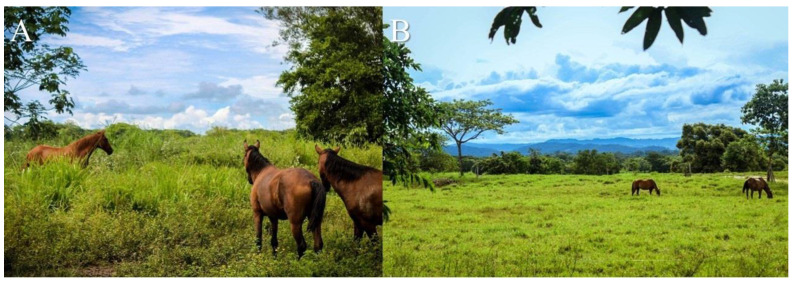
(**A**,**B**) Environmental and feeding conditions through pasture grazing in paddocks of the mares included in the study bred in the Magdalena River meadows of Puerto Berrío (Antioquia, Colombia) (Photos by author, Mauricio Cardona-Garcia).

**Figure 2 animals-14-00760-f002:**
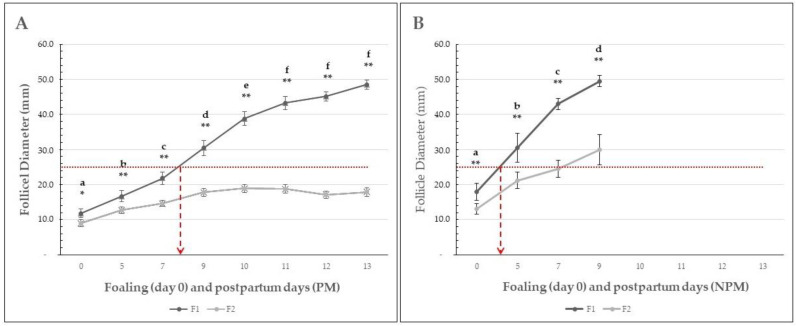
Mean (±SEM) diameter of the first (F1) and second (F2) largest follicle in Colombian paso fino mares that became pregnant (**A**) or non-pregnant (**B**) after breeding by artificial insemination at the foal heat. The dotted red line shows the average size for follicle divergence according to the cut-off value of follicular divergence proposed by [2,6], and the overlapped dashed arrows suggest the postpartum day at follicle divergence. Different letters indicate a significant statistical difference between evaluation days (*p* < 0.01). Asterisks indicate significant statistical differences between F1 and F2. * = *p* < 0.05; ** = *p* < 0.01.

**Figure 3 animals-14-00760-f003:**
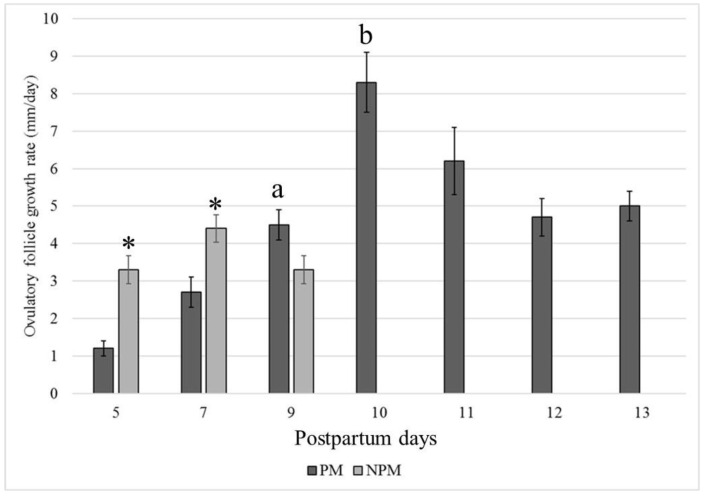
Postpartum day at the highest growth rate (mm/day) of the larger follicle that became the ovulatory follicle in Colombian Paso Fino and crosses with Quarter-horse mares that became pregnant (black color) or not pregnant (gray color) when inseminated at the foal heat (mean values ± standard error). * *p* < 0.05. a, b: *p* = 0.013.

**Figure 4 animals-14-00760-f004:**
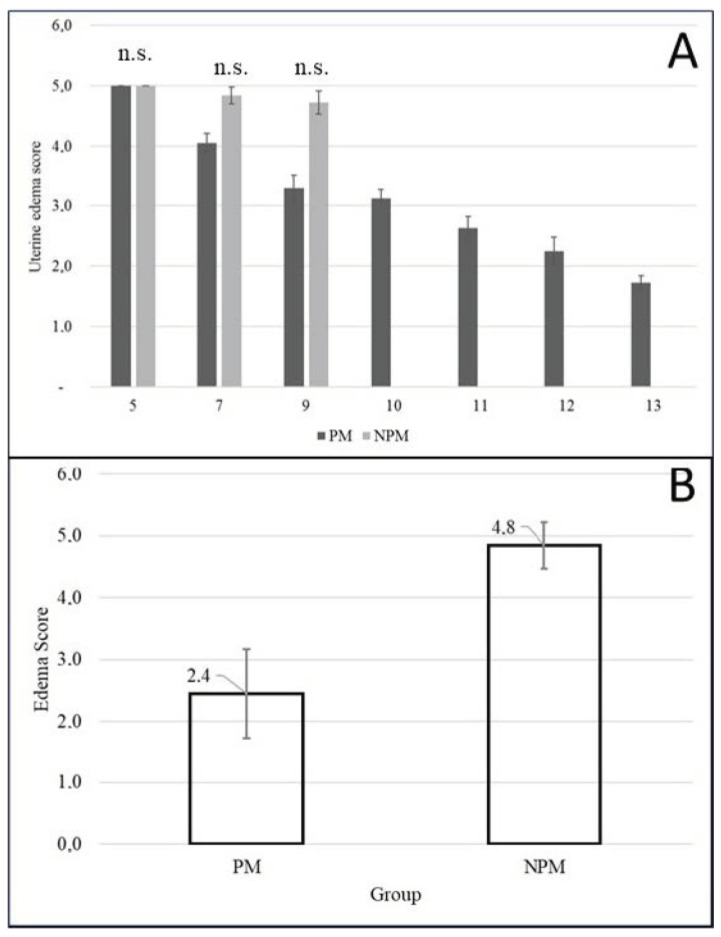
(**A**) Variation in the degree of uterine edema from day five postpartum to the day of the first postpartum ovulation during the foal heat according to pregnancy outcome in Colombian Paso Fino and crosses with Quarter-horse mares (mean ± standard error). n.s. = not statistically significant (Chi-square test; *p* = 0.935). (**B**) Uterine edema score on the day of insemination during the foal heat according to pregnancy outcome in Colombian Paso Fino mares (mean ± SEM). The edema was evaluated by the same investigator by ultrasound exam. PM: pregnant mare group, NPM: non- pregnant mare group.

**Table 1 animals-14-00760-t001:** Epidemiological data of Colombian Paso Fino and crosses with Quarter Horse mares that became pregnant (PM) or non-pregnant (NPM) after insemination at the foal heat.

Mare ID	Breed *	Pregnant *	Age (Years)	Body Weight (Kg)	Foaling	Parity (n)	Gestation Length (Days)
1	CPF	Yes	6	328	12 January 2019	2	328
2	CPF	Yes	6	350	23 January 2019	4	330
3	CPF	Yes	7	349	10 March 2019	3	340
4	CPF	Yes	6	338	18 March 2019	2	339
5	CPF	Yes	9	340	7 March 2019	4	334
6	CPF	Yes	8	330	4 May 2019	3	318
7	CPF	Yes	6	343	23 February 2019	4	332
8	CPF	Yes	5	372	30 May 2019	2	323
9	CPF	Yes	6	345	3 May 2019	5	310
1	CPF	No	8	343	4 March 2019	3	335
2	CPF	No	5	345	16 March 2019	3	332
3	CPF	No	8	353	3 July 2019	4	340
4	CPF	No	7	338	29 July 2019	2	333
5	CPF	No	5	329	7 August 2019	2	328
10	CPF × QH	Yes	7	338	9 January 2020	3	323
11	CPF × QH	Yes	7	362	5 January 2020	3	328
12	CPF × QH	Yes	5	343	28 February 2020	2	331
13	CPF × QH	Yes	8	339	16 February 2020	2	318
14	CPF × QH	Yes	8	328	12 February 2020	4	325
15	CPF × QH	Yes	9	336	28 March 2020	3	338
16	CPF × QH	Yes	7	340	3 March 2020	2	341
17	CPF × QH	Yes	7	330	3 March 2020	1	350
6	CPF × QH	No	8	334	19 January 2020	2	316
7	CPF × QH	No	9	338	29 February 2020	6	335

* No statistically significant differences (*p* > 0.05) were found between pure-bred CPF or their crosses with QH, nor between pregnant and non-pregnant mares. CPF: Pure Colombian Paso Fino mares. CPF × QH: Crossbreed CPF × Quarter Horse mares.

**Table 2 animals-14-00760-t002:** Mean (±SEM) for age, parity, body weight, and gestation length data of mares that became pregnant or non-pregnant after insemination at the foal heat.

Parameter	Pregnant (n = 17)	Non-Pregnant (n = 7)	*p*-Value
Age (years)	6.9 ± 0.3	7.1 ± 0.6	0.665
Parity	2.9 ± 0.3	3.1 ± 0.6	0.627
Body weight (Kg)	341.8 ± 2.8	340.0 ± 3.0	0.743
Gestation length (days)	329.9 ± 2.4	331.3 ± 2.9	0.743

**Table 3 animals-14-00760-t003:** Mean (± SEM) number of follicles on each ovary three days before foaling, at foaling, and diameter of the F1 follicle at foaling in Colombian Paso Fino mares that became pregnant (PM) or non-pregnant (NPM) after insemination at the foal heat.

Time	(n)	Pregnant	(n)	Non-Pregnant	*p*-Value *
Three days before foaling,	34	4.0 ± 0.2	14	3.1 ± 1.2	0.414
Foaling day (day 0)	34	4.2 ± 0.2	14	4.2 ± 0.5	0.477
Largest follicular diameter (F1, day 0)	17	12.1 ± 1.2	7	19.4 ± 2.1	0.004

* Tukey HSD test.

**Table 4 animals-14-00760-t004:** Mean (±SEM) diameter of the first (F1) and second (F2) largest follicles evaluated at foaling and between 5 and 13 days postpartum in Colombian Paso Fino mares that became pregnant (PM) or non-pregnant (NPM) after insemination at the foal heat.

			PM			NPM	
Postpartum Days	Follicle	n *	(Min, Max)	*p*-Value	n *	(Min, Max)	*p*-Value
0	F1	17	12.1 ± 1.2 (7, 23) ^a^		7	19.4 ± 2.1 (15, 37) ^b^	
	F2	16	8.56 ± 0.6 (4, 14)	0.027	7	11.4 ± 0.6 (9, 14)	0.0031
5	F1	17	17.5 ± 1.5 (10, 33) ^a^		7	35.1 ± 2.0 (30, 44) ^b^	
	F2	17	12.1 ± 0.7 (8, 18)	0.0031	7	19.9 ± 1.8 (15, 28)	0.0009
7	F1	17	22.9 ± 1.8 (13, 35) ^a^		6	43.0 ± 1.4 (39, 50) ^b^	
	F2	17	13.9 ± 0.6 (9, 18)	0.00003	6	24.5 ± 2.3 (18, 34)	0.0009
9	F1	17	22.9 ± 1.8 (13, 35)		4	49.5 ± 1.2 (47, 54)	
	F2	17	13.9.0 ± 0.6 (9, 18)	0.00001	4	30.0 ± 3.3 (25, 40)	0.00000
10	F1	17	39.5 ± 1.9 (28, 54)				
	F2	17	18.8 ± 1.1 (11, 29)	0.00001			
11	F1	13	43.1 ± 2.0 (34, 59)				
	F2	13	18.7 ± 1.5 (18, 30)	0.00001			
12	F1	9	45.2 ± 1.8 (38, 53)				
	F2	9	17.1 ± 1.4 (10, 23)	0.00001			
13	F1	7	45.2 ± 1.8 (42, 57)				
	F2	7	17.8 ± 1.9 (10, 23)	0.00001			

* Sample size diminished by the number of mares ovulating every postpartum day. ^a^, ^b^: different letters between 1 diameter from PM and NPM groups mean statistically significant differences (*p* < 0.01).

**Table 5 animals-14-00760-t005:** Mean (±SEM) edema score and diameter of pre-ovulatory follicles in Colombian Paso Fino Mares that became pregnant (PM) or non-pregnant (NPM) after breeding by artificial insemination at the foal heat.

	PM			NPM		
Day	n	Edema Score (Mean)	Diameter (mm) (Min, Max)	n	Edema Score (Mean)	Diameter (mm) (Min, Max)
5				1	5	44 *
7				2	5	42.3 ± 2.4 (42, 44) *
9	3	4	44.3 ± 0.7 (42, 46)	4	5	50.0 ± 1.6 (47, 54) *
10	5	2.75	48.0 ± 3.1 (46, 54) *			
11	4	2.25	49.0 ± 1.5 (40, 59) *			
12	2	1.5	50.0 ± 1.2 (41, 48) *			
13	5	2.0	42.9 ± 1.7 (42, 57) *			

* Indicates mares in which ovulation was confirmed the day after the pre-ovulatory follicle reached its maximum diameter.

**Table 6 animals-14-00760-t006:** The percentage of mares with inflammatory endometrial cytology between 5 and 13 days postpartum, according to the result of foal heat insemination.

Group	Postpartum Days	Cytology *			
		Severe n (%)	Moderate n (%)	Mild n (%)	Normal n (%)
Pregnant Mares	5	17 (100)			
	7	13 (76)	4 (24)		
	9	1 (5.9)	9 (53)	7 (41)	
	10		3 (18)	1 (5.9)	13 (76)
	11		2 (12)	4 (24)	11 (64)
	12			1 (5.9)	16 (94)
	13			1 (5.9)	16 (94)
Not Pregnant Mares	5	7 (100)			
	7	7 (100)			
	9	3 (43)	3 (43)		1 (14)

* The material and methods section showed that the cytology grade was defined using a 5% PMN cut-off point.

## Data Availability

The data presented in this study are available on request from the corresponding author. The data are not publicly available due to [we consider that this information should only be handled by Veterinarians, as it deals with clinical reproductive management in mares].

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
