# Peer review of "Follicular Dynamics and Pregnancy Rates during Foal Heat in Colombian Paso Fino Mares Bred under Permanent Grazing"

_animals, 2024, doi:10.3390/ani14050760_

Round 1
Reviewer 1 Report
Comments and Suggestions for Authors
The manuscript ‘Follicular Dynamics and Pregnancy Rates During Foal Heat in Grazing Paso Fino-Colombiano and Crosses with Quarter Horse Mares’ describes the follicle dynamics during the transition from late pregnancy through parturition and into foal heat in Pasa Fino and Quarter Horse cross mares. The results show that mares that became pregnant at foal heat exhibited slower follicular growth rates and ovulated later while follicle diameter was greater and uterine oedema was lower than in mares that were not pregnant at foal heat.
Specific comments:
L195 – Please clarify the exclusion criteria in terms of point 1. It mentions mares in normal health condition were excluded from the study. Is this referring to mares that were not pregnant?
L148-150 – These lines mention that the mares were supplemented with trace and macro elements, but L205-206 states these mares were not supplemented with trace and macro elements. Please clarify and fix in the text.
L198-205 is repetition from the previous section. Please remove from one of these sections.
L379-380 – non-pregnant and pregnant mares are listed as being the purple and red lines, respectively. However, the lines appear grey and black. Please amend the figure caption to accurately reflect the image. The same colours appear to be used for multiple lines and arrows that do not appear in the figure.
L398 – Please amend the x-axis of the figure, as this contains a spelling error.
L410-411 – Please add the asterix to the caption below the table as this is missing.
General comments:
This is an interesting study and the methods sound. The results are not unexpected given the multitude of other studies in other horse breeds and countries, however, this manuscript provides additional information that is absent in other studies. There is quite a bit of repetition throughout the text which should be removed and simplified given this has been discussed already. Additionally, Table 6 is not mentioned throughout the text. Please make reference to table 6 in the relevant section of the text, or remove if not necessary. Furthermore, please check and amend all figure captions to ensure the information presented matches the image (eg. Colours of groups etc.).
Comments on the Quality of English LanguageThe English language presented in this manuscript can be improved to further enhance readability. Some words are not used as effectively as they could be and grammar can be improved.
Author Response
Cardona et al. Answers to the Reviewer’s comments and suggestions.
Reviewer 1.
“Open Review
(x) I would not like to sign my review report
( ) I would like to sign my review report
Quality of English Language
( ) I am not qualified to assess the quality of English in this paper
( ) English very difficult to understand/incomprehensible
( ) Extensive editing of English language required
( ) Moderate editing of English language required
(x) Minor editing of English language required
( ) English language fine. No issues detected
Yes |
Can be improved |
Must be improved |
Not applicable |
|
Does the introduction provide sufficient background and include all relevant references? |
(x) |
( ) |
( ) |
( ) |
Are all the cited references relevant to the research? |
(x) |
( ) |
( ) |
( ) |
Is the research design appropriate? |
(x) |
( ) |
( ) |
( ) |
Are the methods adequately described? |
(x) |
( ) |
( ) |
( ) |
Are the results clearly presented? |
( ) |
(x) |
( ) |
( ) |
Are the conclusions supported by the results? |
(x) |
( ) |
( ) |
( )” |
Answer: We appreciate the overall concept of the Reviewer on the manuscript. We consider their recommendations of great value to improve the final quality of the manuscript.
“Comments and Suggestions for Authors
The manuscript ‘Follicular Dynamics and Pregnancy Rates During Foal Heat in Grazing Paso Fino-Colombiano and Crosses with Quarter Horse Mares’ describes the follicle dynamics during the transition from late pregnancy through parturition and into foal heat in Pasa Fino and Quarter Horse cross mares. The results show that mares that became pregnant at foal heat exhibited slower follicular growth rates and ovulated later while follicle diameter was greater and uterine oedema was lower than in mares that were not pregnant at foal heat.”
Specific comments:
“L195 – Please clarify the exclusion criteria in terms of point 1. It mentions mares in normal health condition were excluded from the study. Is this referring to mares that were not pregnant?”
Answer: It was a mistake, and as such, it was corrected in the text. Please see line 197.
“L148-150 – These lines mention that the mares were supplemented with trace and macro elements, but L205-206 states these mares were not supplemented with trace and macro elements. Please clarify and fix in the text.”
Answer: It was a mistake, so it was corrected to no provision of concentrated food. Please see line 173.
“L198-205 is repetition from the previous section. Please remove from one of these sections.”
Answer: The repeated section was withdrawn by the suggestion.
“L379-380 – non-pregnant and pregnant mares are listed as being the purple and red lines, respectively. However, the lines appear grey and black. Please amend the figure caption to accurately reflect the image. The same colours appear to be used for multiple lines and arrows that do not appear in the figure.”
Answer: Figure 2 and its legend were replaced with a more detailed figure containing the differences between the first (F1) and second (F2) largest follicle in both PM and NPM. Please see the new Figure 2.
“L398 – Please amend the x-axis of the figure, as this contains a spelling error.”
Answer: It was corrected as suggested. Please see figure 3.
“L410-411 – Please add the asterix to the caption below the table as this is missing.”
Answer: It was corrected as suggested.
General comments:
“This is an interesting study and the methods sound. The results are not unexpected given the multitude of other studies in other horse breeds and countries, however, this manuscript provides additional information that is absent in other studies. “
“There is quite a bit of repetition throughout the text which should be removed and simplified given this has been discussed already.”
Answer: The full text was reviewed and corrected as suggested. A strict grammar review was also performed.
“Additionally, Table 6 is not mentioned throughout the text. Please make reference to table 6 in the relevant section of the text, or remove if not necessary.”
Answer: Table 6 was withdrawn, and this number was assigned to Table 7, and its citation as Table 6 was verified.
“Furthermore, please check and amend all figure captions to ensure the information presented matches the image (eg. Colours of groups etc.).”
Answer: The full text was reviewed and corrected as suggested. A strict grammar review was also performed.
Comments on the Quality of English Language
“The English language presented in this manuscript can be improved to further enhance readability. Some words are not used as effectively as they could be and grammar can be improved.”
Answer: The grammar and writing were cautiously reviewed and corrected. Please see all corrections highlighted in a red color font.
Sincerely
Corresponding Author.
Reviewer 2 Report
Comments and Suggestions for Authors
The idea of the study was interesting, and the obtained results look novel and useful from both theoretical and practical viewpoints. Nevertheless, to my opinion, the manuscript has several weak points, which are to be corrected.
ABSTRACT. It is not clear, why Colombian Paso Fino mares and crosses with Quarter horse, but not the homogenous group has been used. If two groups of horses have been used, where are results of their comparison? The analytical methods should be indicated.
INTRODUCTION should conntain the clear description of the previous related studies, as well as reasons and novelty of the present study. It was done only in Discussion, but not in Introduction.
MATERIALS AND METHODS. Did the two herds contained BOTH Colombian Paso Fino mares and crosses with Quarter horse or one herd contained only Colombian Paso Fino mares and other – only crosses? Why tep herds were selected for the study, if different breeding conditions in these two herds cannot to be excluded?
RESULTS. Why the left and right ovary has been analyzed separately? I believe, the pooling of the data could increase the significance level of the differences between the groups.
DISCUSSION and CONCLUSION contain only repetition of the author’s findings and their simple comparison with the relevant findings of other authors. No true discussion of the causes and mechanisms of the observed differences, the novelty and contribution of the author’s observations to understanding the mechanisms and management of horse’s reproduction has been done. I would suggest to reduce the statements and to increase and to deepen the theoretical and practical parts of the Discussion. Again, the selection of two groups of animals (Colombian Paso Fino mares and crosses) remains not explained, and the differences between these groups (if they were observed) have not been presented and discussed. Otherwise, if the authors did not aimed to perform such comparison, the two groups should be deleted from the title, and the mentions concerning two lines should be minimized in the manuscript..
Author Response
Cardona et al. Answers to the Reviewer’s comments and suggestions.
Reviewer 2.
Open Review
“( ) I would not like to sign my review report
(x) I would like to sign my review report
Quality of English Language
(x) I am not qualified to assess the quality of English in this paper
( ) English very difficult to understand/incomprehensible
( ) Extensive editing of English language required
( ) Moderate editing of English language required
( ) Minor editing of English language required
( ) English language fine. No issues detected
Yes |
Can be improved |
Must be improved |
Not applicable |
|
Does the introduction provide sufficient background and include all relevant references? |
( ) |
( ) |
(x) |
( ) |
Are all the cited references relevant to the research? |
( ) |
(x) |
( ) |
( ) |
Is the research design appropriate? |
(x) |
( ) |
( ) |
( ) |
Are the methods adequately described? |
(x) |
( ) |
( ) |
( ) |
Are the results clearly presented? |
( ) |
(x) |
( ) |
( ) |
Are the conclusions supported by the results? |
(x) |
( ) |
( ) |
( ) |
Comments and Suggestions for Authors
”
“The idea of the study was interesting, and the obtained results look novel and useful from both theoretical and practical viewpoints. Nevertheless, to my opinion, the manuscript has several weak points, which are to be corrected.”
Answer: We appreciate and acknowledge the suggestions of the referee that we accepted, which resulted in a critical improvement of the final quality of the manuscript. A comprehensive review of the English grammar and writing was performed, as can be seen in red color fonts.
“ABSTRACT. It is not clear, why Colombian Paso Fino mares and crosses with Quarter horse, but not the homogenous group has been used. If two groups of horses have been used, where are results of their comparison? The analytical methods should be indicated.”
Answer: It was corrected, and only the genetic background of Colombian Paso Fino mares was left, supported by the lack of statistically significant differences between purebred and crossbred CPF mares in the study.
“INTRODUCTION should conntain the clear description of the previous related studies, as well as reasons and novelty of the present study. It was done only in Discussion, but not in Introduction.”
Answer: In order to avoid repeating information, a concise review was added to the Introduction section as suggested.
“MATERIALS AND METHODS. Did the two herds contained BOTH Colombian Paso Fino mares and crosses with Quarter horse or one herd contained only Colombian Paso Fino mares and other – only crosses? Why tep herds were selected for the study, if different breeding conditions in these two herds cannot to be excluded?”
Answer: In the text, it was stated that due to not statistically significant differences, data was evaluated as CPF mares.
“RESULTS. Why the left and right ovary has been analyzed separately? I believe, the pooling of the data could increase the significance level of the differences between the groups.”
Answer: Pooling data was performed and suggested. Instead of left and right ovaries, the analyses changed, providing data on the first (f1) and second (F2) largest follicles, as can be seen in Figure 2 and Table 4.
“DISCUSSION and CONCLUSION contain only repetition of the author’s findings and their simple comparison with the relevant findings of other authors. No true discussion of the causes and mechanisms of the observed differences, the novelty and contribution of the author’s observations to understanding the mechanisms and management of horse’s reproduction has been done. I would suggest to reduce the statements and to increase and to deepen the theoretical and practical parts of the Discussion. Again, the selection of two groups of animals (Colombian Paso Fino mares and crosses) remains not explained, and the differences between these groups (if they were observed) have not been presented and discussed. Otherwise, if the authors did not aimed to perform such comparison, the two groups should be deleted from the title, and the mentions concerning two lines should be minimized in the manuscript..”
Answer: It was modified as suggested; hopefully, the reviewed version contains the essence of your recommendations.
Respectfully,
The corresponding Author.